# Research on Adaptive Reaction Null Space Planning and Control Strategy Based on VFF–RLS and SSADE–ELM Algorithm for Free-Floating Space Robot

**Xin Ye**, **Zheng-Hong Dong** * and **Jia-Cai Hong**

Space Engineering University, No.1 Bayi Road, Beijing 101416, China; yexintgd@alu.uestc.edu.cn (X.Y.); hongjiacai@sina.com (J.-C.H.)

* Correspondence: dzh.bj@163.com; Tel.: +86-138-1175-7848

**Abstract:** With the increase of on-orbit maintenance and support requirements, the application of a space manipulator is becoming more promising. In actual operation, the strong coupling of the free-floating space robot itself and the unknown disturbance of the contact target caused a major challenge to the robot base posture control. Traditional Reaction Null Space (RNS) motion planning and control methods require the construction of precise dynamic models, which is impossible in reality. In order to solve this problem, this paper proposes a new Adaptive Reaction Null Space (ARNS) path planning and control strategy for the contact of free-floating space robots with unknown targets. The ARNS path planning strategy is constructed by the Variable Forgetting Factor Recursive Least Squares (VFF–RLS) algorithm. At the same time, a robust adaptive control strategy based on the Strategy Self-Adaption Differential Evolution–Extreme Learning Machine (SSADE–ELM) algorithm is proposed to track the dynamic changes of the planned path. The algorithm enables us to intelligently learn and compensate for the unknown disturbance. Then, this paper constructs a robust controller to compensate model uncertainty. A striking feature of the proposed strategy is that it does not require an accurate system model or any information about unknown attributes. This design can dynamically implement RNS path tracking performance. Finally, through simulation and experiment, the proposed algorithm is compared with the existing methods to prove its effectiveness and superiority.

**Keywords:** free-floating space robot; adaptive reaction null space planning; the variable forgetting factor recursive least squares algorithm; the strategy self-adaption differential evolution; extreme learning machine

---

## 1. Introduction

Nowadays, with the increasing frequency of space activities, the impact of spacecraft being hit by space debris has increased. A large amount of space debris has already seriously threatened the safety of on-orbit spacecraft. Especially by a large number of large-scale space debris, they may change the attitude and orbit of the spacecraft and even cause the spacecraft to be completely destroyed [1].

In order to solve such problems, the development of space debris removal technology has become particularly urgent. Among the many active space removal technologies, the technology of space manipulator removal in the orbit has received extensive attention. The ETS–VII of Japan, the Orbital Express of the United States, and the SY-7 of China have successively conducted verification experiments on this technology in space, and are still intensifying their research [2,3].

The related research [4–9] shows that the space robot is in a free-floating state during the process of capturing the target, and the collision from the target contact may lead to a large pulse momentum. The strong dynamic coupling between the spacecraft base and the manipulator may lead to instability of the attitude of the space robot base, which in turn may cause problems such as space robot rollover. The minimum planning of the base attitude disturbance can minimize the disturbance of the manipulator movement on the base posture.

Nenchev [10] proposed an attitude control algorithm for free-floating space robots based on a fixed attitude constrained Jacobian matrix. The algorithm aims to plan the joint movement speed so that the joint angular velocity falls within the null space of the joint inertia matrix, thereby keeping the attitude of the body unchanged. Xie et al. [11] proposed a reactionless coordinated motion planning algorithm with a kinematically redundant manipulator for the dynamic coupling problems of free-floating space manipulators. The numerical results of the planar motion of a 4-links space robotic show that the algorithm is effective. Wei et al. [12] considered the attitude reactionless control and the vibration suppression in the meantime to reduce the risk in grasping operation. The simulation results indicate that, by using the optimal control for the vibration suppression in the attitude RNS, the vibration of the manipulator could be alleviated significantly and the base attitude could also almost be undisturbed in the meantime during the whole grasping procedure.

The space manipulator equipped with a flexible mechanism can better achieve the collision force buffering and unloading during the contact with the target. However, the contact between the manipulator and the target will stimulate the flexible manipulator vibration and cause substantial disturbance to the base attitude, due to the momentum conservation in space. The flexible manipulator vibration and the impulse to the base attitude are critical to the contacting safety and the performance. The uncertain properties from the captured unknown objects such as space debris can cause the conventional no-disturbance planning methods to be invalid. To solve this problem, the Adaptive Reaction Null Space (ARNS) method [13,14] with dynamic changing planning paths for manipulators during capturing of an unknown object was proposed and introduced to realize the minimum disturbance to the spacecraft body attitude. In this method, the unknown part of the target can be obtained online via the Recursive Least Square (RLS) algorithm, and compensation can be made in the form of a time-variable matrix for the path planning equation of the manipulator. Xu et al. [15] proposed an adaptive control algorithm based on Reaction Null Space (RNS) for free-floating robots with uncertain kinematics and dynamics, which realizes base attitude adjustment and continuous path tracking of end effectors. Similar schemes should acquire accurate information of the captured object and bring about high sensor requirements and a delay control issue. Lu and Jia [16] made the spacecraft attitude regulation error and the end–effector pose tracking error meeting the respective prescribed performance requirements, converging to zero by the ARNS and the prescribed performance functions. Thus, the coupling effects of the free-floating robots are overcome. Zhang [17] studied the target dynamics parameter identification based on ARNS planning and used the RLS algorithm to update the system in real-time. This method does not need the iteration of unknown dynamic parameters of non-cooperative targets, and the base parameter identification is realized. The attitude disturbance is minimal and the adaptive control can be separated from the dynamic parameter identification. Jiao et al. [18] derived a dual-arm space robot model and proposed an ARNS motion control approach to satisfy the principal objective of maintaining a minimum disturbance to the base in the post-capture of a non-cooperative target. They developed a new ARNS control for a dual-arm space robot system. The adaptive algorithm is developed based on the momentum conservation of the system and the RLS algorithm is employed for parameter adaptation. The simulation results are presented to demonstrate the effectiveness of the proposed approach.

The traditional RLS algorithm has data saturation phenomenon [19]. When the space robot contacts an unknown target, the recursive algorithm cannot be directly used because the system parameters change with time, thereby reducing the performance of the ARNS algorithm path planning. In order to adapt to the case of time-varying parameters, the forgetting factor can be added to reduce

the weight of the old data and increase the weight of the new data, thereby increasing the adaptability of the ARNS algorithm to the time-varying system [17]. In addition, the sliding-window RLS (SW–RLS) algorithm eliminates the saturation of the RLS algorithm by retaining a finite data length [20]. To realize the reactionless motion for a space robot while capturing an unknown object, not only the ARNS path planning algorithm based on dynamic characteristics, but also the space robot attitude tracking control algorithm, capable of effectively tracking the planned path, is needed. However, when the space robot itself has uncertainty and the target state is unknown, in order to ensure the stability of the system attitude and reduce the influence of tracking error, the tracking control strategy of the system needs to learn the unknown disturbance and compensate through the adaptive strategy, while robust strategy culls model uncertainty, such as in Reference [21]. However, traditional adaptive strategies, such as Back-Propagation neural network [22], radial basis function neural network [23], and fuzzy algorithms [24,25], are not processing fast enough to handle disturbance with bursty properties. We need a faster learning and compensating method for the abrupt unknown properties.

The Extreme Learning Machine (ELM) [26,27] algorithm was developed for its extremely fast system uncertainty learning and cognizing performance [28–30]. However, the random input feature settings (input weight, hidden layer bias) of conventional ELM networks could weaken the network performance to some extent, because the pre-setting input feature owns the possibility of staying away from its optimal value.

Given the problem, some ameliorated methods by optimizing the input feature of the ELM algorithm have been proposed. Figueiredo et al. [31] studied the effects of eight different topologies on the performance of Particle Swarm Optimization-ELM (PSO–ELM). The results show that there is no optimal topology suitable for all problems, but according to the root mean square error from the experimental results, the global topologies perform better. Zhang et al. [32] proposed a Memetic Algorithm Based-ELM (M–ELM) algorithm. M–ELM integrates individual heuristic searches into the global optimization framework of the population to automatically learn the optimal network parameters, and effectively overcomes the problem of premature convergence. Cao et al. [33] proposed a Self-adaptive Evolutionary (SaE–ELM) algorithm that not only adaptively adjusts the crossover probability and scaling factor, but also automatically selects the best mutation strategy from the four-seed generation strategy. The convergence of the algorithm is improved. Tang et al. [34] proposed a Self-adaptive Differential Evolutionary–Weighted ELM (SDE–WELM) algorithm, which uses an adaptive differential evolution algorithm to optimize the parameters of hidden layer neurons and the weight of training samples, improving the accuracy of unbalanced data classification. Chu et al. [20] proposed a robust adaptive control strategy, using PSO–ELM algorithm to dynamically learn the reactionless path of space robot planning, and compensate for the unknown characteristics in the form of constructive adaptive control. Using the ELM algorithm to learn fast, the mutation characteristics of the space robot system are processed, and the input characteristics of the ELM are optimized by the PSO algorithm, which improves the ELM performance.

Considering the uncertainty of the space robot system and unknown external disturbance, and in order to meet the requirements of space robot for ARNS planning and manipulator tracking control with faster adaptive performance and higher tracking accuracy, the strategy uses the Variable Forgetting Factor Recursive Least Squares (VFF–RLS) algorithm to identify and compensate for the unknown characteristics of the system, and avoids the saturation phenomenon when dealing with time-varying elements in the planning process. At the same time, the ELM algorithm is improved by the Strategy Self-Adaptation Differential Evolution (SSADE) algorithm to effectively compensate for the unknown interference in the tracking control process. Based on this, this paper proposes an ARNS path planning strategy based on the VFF–RLS algorithm, and a tracking control algorithm based on SSADE–ELM algorithm.

The work carried out in this paper is as follows: In Section 2, the dynamic model of the space robot is established. In Section 3, the planning Strategy and Adaptive robust control algorithm constructed by the ARNS Planning Strategy based on VFF–RLS, adaptive robust control algorithm

based on SSADE–ELM, and robust control algorithm are elaborated. The simulation results are shown in Section 4. The experimental results are shown in Section 5. The discussion is shown in Section 5. Finally, the conclusions are provided in Section 6.

## 2. Dynamic Model of the Space Robot

The simplified model of the space robot designed in this paper is shown in Figure 1. Where $\Sigma_I$ is the inertial frame; $\Sigma_i(i-1,2,\ldots,n)$ is the *i*th body frame and $\Sigma_0$ is the base frame, where $b_0$ is a vector pointing from center of the base to the centroid of the first joint. $r_0$, $v_0$, $\omega_0$ are the position vector of the centroid of the base, the velocity and angular velocity of the base, respectively. $a_i$ is a vector pointing from center of the *i*th joint to the centroid of the *i*th body and $b_i$ is a vector pointing from the centroid of the ith body to the center of the $(i + 1)$th joint. $r_{0g}$ is the vector from origin of the mass center of the base to the mass center of the system. $r_g$ is the position vector of the whole system with respect to inertial frame.

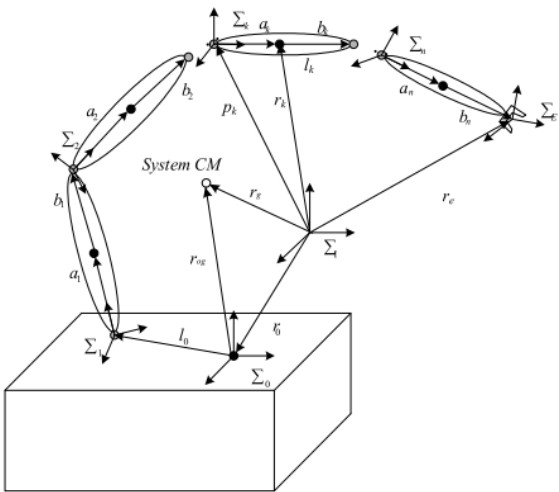

**Figure 1.** The simplified model of space manipulator.

Considering that the space robot is in a free-floating state during the contact with the target and the post-contact stabilization process, the complex system composed of the space robot and the target is not subjected to any external force, and the complex system maintains the angular momentum and the linear momentum conservation. The only disturbance in the system is due to internal factors such as joint friction, measurement noise, and uncertainty of the target parameters. The Lagrange equation can be used to construct the dynamic model of a space robot as in Equation (1):

$$M_t\ddot{q}+C_t\dot{q}+\tau_k=\tau_e; \Delta M=M_t-M, \Delta C=C_t-C \tag{1}$$

Equation (1) represents the complex dynamics model in which the dynamic parameters of the system are changed based on the dynamic parameters of the original space robot after the space robot captures the unknown target to form the complex. $q$ is the generalized state variable of joint angular displacement. $M$ and $C$, respectively, represent the system equivalent moment of inertia and equivalent friction coefficient of the joint of the space manipulator. $M_t$ and $C_t$ represent the actual moment of inertia and the actual frictional damping coefficient of the joint of the space manipulator joint, respectively. $\Delta$ represents an unknown system model parameter perturbation. $\tau_k$ is the joint actuation torque. $\tau_e$ is an unknown bounded external disturbance. The model parameter uncertainty

in the system dynamics model in Equation (1), the unknown friction torque, and the unknown external disturbance can be expressed as a form of composite interference, as shown in Equation (2).

$$M\ddot{q} + C\dot{q} + \tau_k = d$$
$$d = \tau_e - \Delta M\ddot{q} - \Delta C\dot{q}$$

(2)

where $M = diag\{M_{11}, \ldots M_{ii} \ldots, M_{nn}\}, \Delta M = diag\{\Delta M_{11}, \ldots \Delta M_{ii} \ldots, \Delta M_{nn}\}$ are constant coefficient diagonal matrices. So, Equation (2) can be rewritten as

$$\ddot{q} = M^{-1}(d - C\dot{q} - \tau_k) = M^{-1}(d - C\dot{q}) - M^{-1}\tau_k$$

(3)

Setting $d' = M^{-1}(d - C\dot{q})$, where $d' = [d'_1, d'_2, \ldots, d'_n]$. Since $M$ is a diagonal reversible matrix, it can be proved that Equation (3) is decoupable. The angular displacement of the above equation after decoupling has the following form.

$$\ddot{q}_i = d'_i - M_{ii}^{-1}\tau_{ki}$$

(4)

Let $q_d$ be the generalized angular displacement of the joint desired. Then there are joint angle error and angular velocity error, as shown in the following equation.

$$e = q - q_d; \dot{e} = \dot{q} - \dot{q}_d$$

(5)

The control variable of the joint angle error system of the space robot can be expressed as:

$$r = \dot{e} + \Lambda e$$

(6)

where $\Lambda$ is a positive diagonal matrix. Since $r$ tends to zero, the reference output of the joint angular displacement can be expressed as:

$$\dot{q}_r = \dot{q}_d + \Lambda e$$

(7)

Bringing Equation (7) into Equation (1) gives:

$$
\begin{aligned}
&M(\ddot{q}_d + \Lambda\dot{e}) + C(\dot{q}_d + \Lambda e) + \tau_k - d \\
&= M(\ddot{q}_d + \Lambda\dot{e}) + C(\dot{q}_d + \Lambda e) - (M\ddot{q} + C\dot{q} - d) - d \\
&= M(\ddot{q}_d + \Lambda\dot{e} - \ddot{q}) + C(\dot{q}_d + \Lambda e - \dot{q}) \\
&= M\dot{r} + Cr
\end{aligned}
$$

(8)

## 3. Adaptive Reaction Null Space (ARNS) Planning Strategy and Adaptive Robust Control Algorithm

### 3.1. Adaptive Reaction Null Space (ARNS) Planning Strategy Based on Variable Forgetting Factor Recursive Least Squares (VFF–RLS)

After the space robot captures the target, the motion of the robot arm may cause the base posture to be greatly disturbed. This section uses the ARNS planning to solve the problem that the complex system has the least disturbance to the base during the motion. The minimum perturbation to the base attitude is achieved by planning the reaction null space joint angular velocity with unknown target kinetic parameters. This section uses the least squares method to complete the adaptive update based on the smallest error prediction, since the conventional least squares method has saturation and is not suitable for the case of time-varying system parameters. Therefore, the variable forgetting factor is used to improve the least squares method, and the VFF–RLS algorithm is constructed to achieve ARNS planning.

The space robot in this paper is in a free-floating state, and its manipulator has redundancy. The centroid position vector of each arm of the space robot complex system is as follows.

$$r_i = r_0 + b_0 + \sum_{k=1}^{i-1}(a_k + b_k) + a_i \tag{9}$$

Setting $J_{Ri} = [Z_1, Z_2, \ldots, Z_i, 0, \ldots, 0]$ and $J_{Ti} = [Z_1 \times (r_i - p_1), Z_2 \times (r_i - p_2), \ldots, Z_i \times (r_i - p_i), 0, \ldots, 0]$. The linear velocity and angular velocity at the centroid of each boom are as follows.

$$v_i = \dot{r}_i = v_0 + \omega_0 \times (r_i - r_0) + J_{Ti}\dot{q} \tag{10}$$

$$\omega_i = \omega_0 + J_{Ri}\dot{q} \tag{11}$$

The linear momentum equation of the system is:

$$P = \sum_{i=0}^{n}(m_i \dot{r}_i) = \sum_{i=0}^{n}(m_i \dot{r}_i) \tag{12}$$

Substituting Equation (10) into Equation (12) can obtained that:

$$P = Mv_0 + \left(M\tilde{r}^T{}_{0g}\right)\omega_0 + J_{T\omega}\dot{q} = \begin{bmatrix} ME & M\tilde{r}^T{}_{0g} \end{bmatrix}\begin{bmatrix} v_0 \\ \omega_0 \end{bmatrix} + J_{T\omega}\dot{q} \tag{13}$$

where $r_{0g} = r_g - r_0$, $J_{T\omega} = \sum_{i=1}^{n}(m_i J_{Ti})$.

Similarly, the system satisfies the following angular momentum conservation equation:

$$L = \sum_{i=0}^{n}\left(I_i\omega_i + m_i r_i \times \dot{r}_i\right) \tag{14}$$

Substituting Equations (10) and (11) into Equation (14) yields the following equation:

$$\begin{aligned} L &= I_0\omega_0 + \sum_{i=1}^{n} I_i\left(\omega_0 + J_{Ri}\dot{q}\right) + \sum_{i=1}^{n} m_i\tilde{r}_i\left(v_0 - \tilde{r}^T{}_{0i}\omega_0 + J_{Ti}\dot{q}\right) \\ &= \begin{bmatrix} M\tilde{r}_{0g} & I_\omega \end{bmatrix}\begin{bmatrix} v_0 \\ \omega_0 \end{bmatrix} + I_{\omega\phi}\dot{q} \end{aligned} \tag{15}$$

where $I_\omega = \sum_{i=1}^{n}\left(I_i - m_i\tilde{r}_i^T\tilde{r}_{0i}\right) + I_0$, $I_{\omega\phi} = \sum_{i=1}^{n}(I_i J_{Ri} + m_i\tilde{r}_i J_{Ti})$.

The angular momentum of the multi-body system set as $L_g$ is calculated around the centroid of the whole space robot. The angular momentum conservation equation can be rewritten as:

$$L = L_g + r_g \times P \tag{16}$$

The angular momentum and linear momentum equations of the space robot after contact with the target can be obtained. In conjunction with Equations (13) and (15), the momentum conservation equation can be expressed as in Equation (17), where $P$ is the complex system linear momentum, and $L$ is the angular momentum of the composite system.

$$\begin{bmatrix} P \\ L \end{bmatrix} = \begin{bmatrix} ME & M\tilde{r}^T{}_{0g} \\ 0 & H_\omega \end{bmatrix}\begin{bmatrix} v_0 \\ \omega_0 \end{bmatrix} + \begin{bmatrix} J_{T\omega} \\ H_{\omega\phi} \end{bmatrix}\dot{q} + \begin{bmatrix} 0 \\ r_g \times P \end{bmatrix} \tag{17}$$

where $H_\omega = \sum\limits_{i=1}^{n}\left(I_i + m_i\widetilde{r}_{gi}^T\widetilde{r}_{0i}\right) + I_0$, $H_{\omega\phi} = \sum\limits_{i=1}^{n}\left(I_iJ_{Ri} + m_i\widetilde{r}_{gi}J_{Ti}\right)$, $M$ is the total mass of the space robot target complex system.

The angular momentum equation of the space robot complex system obtained by Equation (17) is as follows.

$$L = H_\omega\omega_0 + H_{\omega\phi}\dot{q} + r_g \times P \tag{18}$$

When the parameters of the complex system are known, the RNS-based composite system manipulator joint angular motion plan is:

$$\dot{q}_{d|RNS} = H_{\omega\phi}^{+}\left(L - r_g \times P\right) + \left(E - H_{\omega\phi}^{+}H_{\omega\phi}\right)\dot{\xi} \tag{19}$$

where $E - H_{\omega\phi}^{+}H_{\omega\phi}$ is the null space projection of the coupling matrix $H_{\omega\phi}$. $H_{\omega\phi}^{+}$ is the generalized inverse matrix of $H_{\omega\phi}$. $\dot{\xi} \in R^n$ is any non-zero vector. However, since the target angular momentum is an unknown variable, $H_\omega$ and $H_{\omega\phi}$ in Equation (19) cannot be accurately obtained, and the estimated values are used, which are denoted as $\widetilde{H}_\omega$ and $\widetilde{H}_{\omega\phi}$. Therefore, using the ARNS motion planning algorithm to adjust the coefficient matrix online, the accurate performance of the complex system can be obtained.

$$\dot{q} = \widetilde{H}_{\omega\phi}^{+}\left(L - r_g \times P\right) - \widetilde{H}_{\omega\phi}^{+}\widetilde{H}_\omega\omega_0 + \left(E - \widetilde{H}_{\omega\phi}^{+}\widetilde{H}_{\omega\phi}\right)\dot{\xi} \tag{20}$$

Setting $y = \dot{q}_{d|RNS} - \dot{q}$, then bringing in Equation (20) can obtain Equation (21) as follows.

$$
\begin{aligned}
y = \ \dot{q}_{d|RNS} - \dot{q} &= \left(H_{\omega\phi}^{+} - \hat{H}_{\omega\phi}^{+}\right)\left(L - r_g \times P\right) + \hat{H}_{\omega\phi}^{+}\hat{H}_\omega\omega_0 + \left(\hat{H}_{\omega\phi}^{+}\hat{H}_{\omega\phi} - H_{\omega\phi}^{+}H_{\omega\phi}\right)\dot{\xi} \\
&= K_1 + K_2\omega_0 + K_3\dot{\xi} \\
&= \begin{bmatrix} K_1 & K_2 & K_3 \end{bmatrix} \begin{bmatrix} 1 \\ \omega_0 \\ \dot{\xi} \end{bmatrix}
\end{aligned} \tag{21}
$$

where $K_1 = \left(H_{\omega\phi}^{+} - \hat{H}_{\omega\phi}^{+}\right)\left(L - r_g \times P\right)$; $K_2 = \hat{H}_{\omega\phi}^{+}\hat{H}_\omega$; $K_3 = \left(\hat{H}_{\omega\phi}^{+}\hat{H}_{\omega\phi} - H_{\omega\phi}^{+}H_{\omega\phi}\right)$.

Define the error function $\varepsilon$ is as follows.

$$\varepsilon = y - \left(\hat{K}_1 + \hat{K}_2\omega_0 + \hat{K}_3\dot{\xi}\right) \tag{22}$$

Since the parameters $K_1$, $K_2$, and $K_3$ contain uncertain parameters of non-cooperative targets, in order to ensure that the angular velocity of the pedestal converges to zero, an algorithm is needed to continuously update the values of parameters $K_1$, $K_2$, and $K_3$, so that the error function $\varepsilon$ is minimal.

In order to ensure that the unknown target has the least disturbance to the base attitude, the parameters $K_1$, $K_2$, and $K_3$ are automatically updated online according to Equation (15) using the VFF–RLS algorithm to achieve the effect of adaptively updating the RNS velocity of the joint. Equation (15) can be written as a standard regression equation as follows.

$$y^T = \Phi W \tag{23}$$

where $\Phi = \begin{bmatrix} 1 & \omega_0 & \dot{\xi} \end{bmatrix}^T$, $W = \begin{bmatrix} K_1 & K_2 & K_3 \end{bmatrix}^T$. According to the VFF–RLS algorithm, Equation (20) is the update equation of $W$.

$$\hat{W}_{(k)} = \hat{W}_{(k-1)} + N_{(k)}\varepsilon_1 \tag{24}$$

where

$$\varepsilon_1 = y_{(k)}^T - \Phi_{(k)} \hat{W}_{(k)}$$
$$N_{(k)} = \frac{Q_{(k-1)} \Phi_{(k)}}{\lambda_{(k)} E + \Phi_{(k)}^T Q_{(k-1)} \Phi_{(k)}}$$
$$Q_{(k)} = \lambda_{(k)}^{-1} E \left[ E - N_{(k)} \Phi_{(k)}^T \right] Q_{(k-1)}$$
(25)

If $\lambda_{(k+1)}$ is a constant, it cannot achieve good stability and tracking capabilities at the same time. In fact, when $\lambda_{(k+1)}$ is small, the algorithm has fast convergence rate but poor stationarity, and when $\lambda_{(k+1)}$ is large, the algorithm has converse performance. It is not an easy work to find an appropriate constant. Therefore, a new form of changing rule is introduced, where the convergence rate is fast in the beginning phase and has good stability in the subsequent phase. In this paper, the following changing rule is adopted.

$$\lambda(k) = \lambda_{max} - \sigma_1 e^{-\sigma_2 k}$$
(26)

where $\sigma_1$ and $\sigma_2$ are two coefficients that control the changing manner of forgetting factor. In the above formula, $\hat{W}_{(k)}$ is an estimate of the kth generation of $W_{(k)}$. $\varepsilon_1$ represents the a priori residual. $Q_{(k)}$ is the inverse of the autocorrelation matrix. $Q_{(0)} = \delta E, \delta > 1$. Here $\delta = 1.4$. The forgetting factor is close to 1 to ensure that the ARNS planning algorithm is stable during the post-capture motion. So, take $\sigma_1 = 0.1, \sigma_2 = 0.04, \lambda_{max} = 0.99$. When $\hat{W}$ is updated, the expected RNS joint angular velocity can be expressed as:

$$\dot{q}_{d|RNS} = \dot{q} + \hat{W}^T \begin{bmatrix} 1 \\ \omega_0 \\ \dot{\xi} \end{bmatrix}$$
(27)

Figure 2 shows the ARNS path planning algorithm flow.

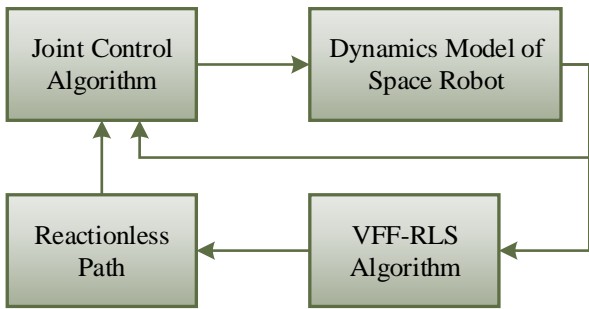

**Figure 2.** The Adaptive Reaction Null Space (ARNS) path planning algorithm flow diagram.

*3.2. Adaptive Control Algorithm Based on Strategy Self-Adaptation Differential Evolution–Extreme Learning Machine (SSADE–ELM)*

The space robot dynamics model constructed in this paper is a nonlinear system with unknown disturbances. The nonlinear system has parameter perturbation, while adaptive control and robust control are the main means to cancel the parameter perturbation of nonlinear systems. This section presents an adaptive robust control algorithm based on SSADE–ELM. The diagram of adaptive control algorithm based on SSADE–ELM is shown in Figure 3. The control law is as follows:

$$\tau_{ki} = M_{ii}(\tau_{ai} + \tau_{ci} + \tau_{PDi})$$
(28)

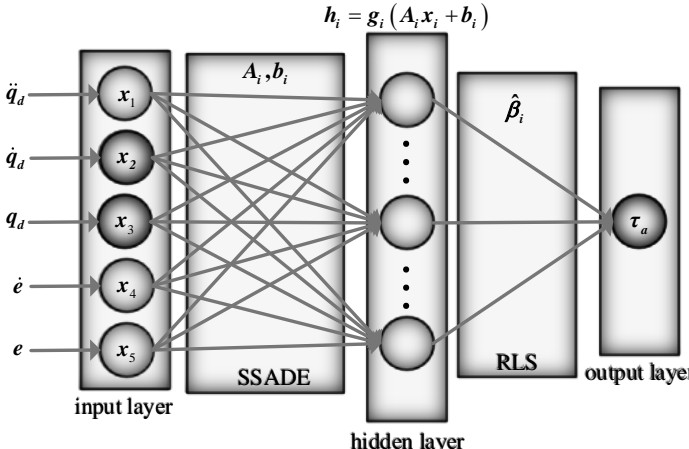

**Figure 3.** Diagram of adaptive control algorithm based on Strategy Self-Adaptation Differential Evolution–Extreme Learning Machine (SSADE–ELM).

$\tau_{ai}$ represents an adaptive controller for compensating for uncertain or unknown disturbance present in the system. $\tau_{ci}$ represents a robust controller that compensates for model uncertainty. $\tau_{PDi}$ represents the feedback controller, which is the PD controller. According to Equations (4) and (28), Equation (29) can be obtained:

$$d'_i = \ddot{q}_i - (\tau_{ai} + \tau_{ci} + \tau_{PDi}) \tag{29}$$

Since the ELM algorithm has a strong approximation ability, it can be applied to a controller that compensates for unknown disturbance of a space robot without requiring prior knowledge. This method enables real-time control of the robot and accurately tracks the planned trajectory. The input vector of the ELM network is $x_i$, which is defined as shown in the following equation.

$$x_i = \begin{bmatrix} \ddot{q}_{di} & \dot{q}_{di} & q_{di} & e_i & \dot{e}_i \end{bmatrix} \tag{30}$$

Therefore, the output of the hidden layer node can be defined as:

$$h_i = g_i(A_i x_i + b_i) \tag{31}$$

where $g_i$ is the activation function. Here it uses the sigmoid function. $A_i$ is the input weight indicating the connection between the hidden layer neurons and the input layer neurons, and $b_i$ is the hidden layer bias. The adaptive compensation controller $\tau_a$ is constructed using the ELM network as follows:

$$\tau_{ai} = h_i \beta_i \tag{32}$$

where $\beta_i = \begin{bmatrix} \beta_{i1}, \beta_{i2}, \ldots, \beta_{ij} \end{bmatrix} (j=1,2,\ldots,l)$ is a weight matrix between the output layer and the hidden layer, which can be solved by the RLS algorithm.

$$\beta_i = h_i^+ \tau_{ai} \tag{33}$$

where $h_i^+$ is the Moore–Penrose generalized inverse of $h_i$. The adaptive law of $\hat{\beta}_i$ can be expressed as:

$$\dot{\hat{\beta}}_i = \mu_i h_i r_i \tag{34}$$

where $\dot{\hat{\beta}}_i = \begin{bmatrix} \dot{\hat{\beta}}_{i1}, \dot{\hat{\beta}}_{i2}, \ldots, \dot{\hat{\beta}}_{ij} \end{bmatrix}$. $\mu_i = \begin{bmatrix} \mu_{i1}, \mu_{i2}, \ldots, \mu_{ij} \end{bmatrix}$, $(j=1,2,\ldots,l)$ is the step adjustment factor. $\mu_{ij} = 1$. $l$ is the number of hidden layer neurons. According to Equation (32), Equation (35) can be obtained:

$$\hat{\tau}_{ai} = h_i \hat{\beta}_i \tag{35}$$



The improved SSADE algorithm is introduced into the ELM algorithm. The differential variation and crossover operator of the algorithm can be used to search for the optimal input weight $A_i$ and hidden layer bias $b_i$ according to the dynamic adjustment of the whole population, thus obtaining a more compact network structure. This can avoid excessive random neurons and cause the hidden layer to have no sparsity and adjustment ability, which affects the generalization ability and stability of the network.

Compared with the traditional differential evolution algorithm [35], the algorithm adopts different mutation strategies in different periods of population evolution, which improves the convergence speed and convergence precision of the algorithm. The implementation steps shown in Figure 4 are as follows:

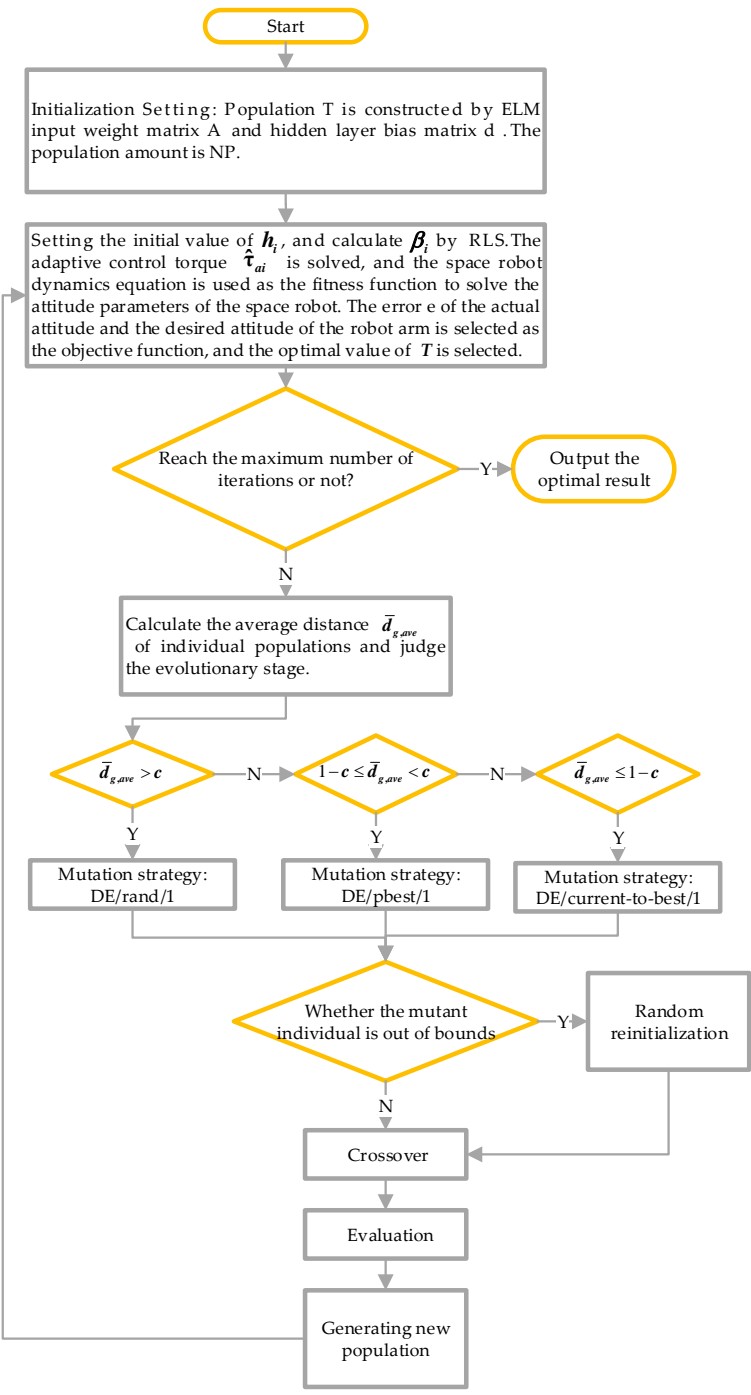

**Figure 4.** The improved Strategy Self-Adaptation Differential Evolution (SSADE) algorithm flow chart.

Step 1: Initialization. Set the ELM hidden layer unit number *l* and the excitation function g(x). The population $T = \left[ a_{r,g}^{11}, a_{r,g}^{12}, \cdots, a_{r,g}^{1n}, \cdots, a_{r,g}^{l1}, a_{r,g}^{l2}, \cdots, a_{r,g}^{ln}, d_{r,g}^{1}, d_{r,g}^{2}, \ldots, d_{r,g}^{l} \right]$ is constructed from the input weight matrix and the implicit layer bias matrix of the ELM, and g represents the number of iterations. Initialize Np vectors with dimension of D ($l \times (n + 1)$). For each individual population, the range of each dimension is [−1, 1]. The hidden layer output matrix $h_i$ is calculated according to Equation (31), and the output weight $\beta_i$ is calculated according to Equation (33).

Step 2: Mutation. Aiming at the problem of mutation strategy selection based on the DE algorithm, based on the staged idea, this paper proposes a phased strategy adaptive differential evolution algorithm. The proposed algorithm includes two parts: the evolutionary stage estimation based on the population congestion degree and the adaptive selection of the mutation strategy.

1. The evolutionary stage estimation based on the population congestion degree.

Referring to the idea of crowded density estimation in multi-objective differential evolution algorithm, the judgment basis of the evolutionary stage is set as the ratio of the average distance $d_{g,ave}$ to the maximum distance $d_{max}$ between each individual and the optimal individual (the minimum objective function value) in the current population. This average distance is as follows:

$$d_{g,ave} = \frac{\sum\limits_{r=1}^{Np} \sqrt{\sum\limits_{j=1}^{D} \left( t_{r,g}^{j} - t_{best,g}^{j} \right)^2}}{Np} \tag{36}$$

where $t_{r,g}^{j}$ is the current individual in the *g*th generation population. $t_{best,g}^{j}$ is the best individual in the *g*th generation population. Compare the average distance of each generation with $d_{max}$. If the average distance of the current generation is greater than $d_{max}$, then replace $d_{max}$. Since the population eventually converges to a point, the minimum value of the average distance is $d_{min} = 0$. Then, the average distance of each generation is normalized according to the maximum and minimum values of the average distance, i.e.,

$$\overline{d}_{g,ave} = \frac{d_{g,ave} - d_{min}}{d_{max} - d_{min}} = \frac{d_{g,ave}}{d_{max}} \tag{37}$$

Finally, based on the $\overline{d}_{g,ave}$ of each generation, estimate the stage of the current population.

$$\begin{cases} S_1, if \quad \overline{d}_{g,ave} > c \\ S_2, if \, 1 - c \leq \overline{d}_{g,ave} < c \\ S_3, otherwise \end{cases} \tag{38}$$

where $S_1$, $S_2$, and $S_3$ denote the first, second, and third stages, respectively, and *c* is the stage control factor.

2. The adaptive selection of the mutation strategy.

In order to improve the search efficiency of the algorithm, while maintaining the diversity of the population, avoiding the algorithm falling into local optimum and premature convergence, different strategies are designed to mutate according to the characteristics of different stages.

In the first stage $S_1$, the whole population is relatively dispersed, and all individuals are committed to searching for promising sub-regions. At this time, the population diversity should be maintained to ensure that as many regions as possible are searched. Therefore, in the first phase, the DE/rand/1mutation strategy is adopted.

$$V_{i,g} = T_{r1,g}(t) + F\left( T_{r2,g} - T_{r3,g} \right) \tag{39}$$

where $V_{i,g}$ indicates a variant individual. $T_{r1,g}$, $T_{r2,g}$, and $T_{r3,g}$ are randomly selected individuals. $r_1 \neq r_2 \neq r_3 \in [1, Np]$. F is a contraction factor, which obeys Cauchy distribution C (0.5, 0.1).

In the second stage, $S_2$, the algorithm begins to detect promising regions that have been searched and searches further for more promising regions. At this point, the algorithm needs to perform both global detection and local search. Therefore, in order to balance the relationship between population diversity and convergence rate, the DE/pbest/1 mutation strategy is adopted.

$$V_{i,g} = T_{pbest,g} + F\left(T_{r1,g} - T_{r2,g}\right) \tag{40}$$

where $T_{pbest,g}$ is the optimal individual among the historical optimal solutions before the g-generation of the target individual $T_{i,g}$. $T_{r1,g}$, and $T_{r2,g}$ are randomly selected individuals. $F$ is a contraction factor. Through the collaboration of historically optimal individuals and other individuals to guide the mutation, not only can the population diversity be maintained, but also the algorithm can search for as many regions as possible, and the local search ability of the algorithm can be improved, thereby accelerating the convergence speed.

In the third stage, $S_3$, all individuals may be clustered into a certain area. The algorithm is dedicated to searching for the optimal solution in the area. Therefore, in order to speed up the convergence, the DE/current-to-best/1 mutation strategy is adopted.

$$V_{i,g} = T_{i,g} + F\left(T_{best,g} - T_{i,g}\right) + \lambda\left(T_{r1.g} - T_{r2,g}\right) \tag{41}$$

where $T_{best,g}$ is the best individual in the entire population. $T_{r1,g}$ and $T_{r2,g}$ are two different randomly selected individuals in the population other than $T_{best,g}$. By using the information of the current optimal individual to guide the mutation, the algorithm converges quickly.

Boundary operation: The mutation individuals generated by the mutation operator may exceed the allowable range of the boundary. If $V_{i,g}$ is out of bounds, the individual is randomly generated within its range of values as follows:

$$V_{r,g} = rand(0,1)\left(t_r^U - t_r^L\right) + t_r^L \tag{42}$$

where $t_r^U$ and $t_r^L$ are the upper and lower bounds of $t_r$, respectively. Rand (0, 1) represents a random fraction between (0, 1) and obeys a uniformly distributed.

Step 3: Crossover. The crossover operation increases the diversity of the group is as follows.

$$T_{trial,r,g} = \begin{cases} V_{r,g}, & if \quad R \leq CR \\ T_{r,g}, & otherwise \end{cases} \tag{43}$$

where $CR$ obeys a normal distribution N (0.5, 0.3). $R \in (0,1)$ obeys uniformly distributed.

Step 4: Selection. According to the planning of the minimum base disturbance of the free-floating space robot, the dynamic equation of the space robot is selected as the fitness function. $f = e$ is the objective function. That is, the attitude parameters of the space robot need to be close to the desired attitude parameters obtained through planning. When the value of $f$ decreases, the value of $T$ approaches the optimal. Calculate the objective function values $f\left(t_{trial,r,g}\right)$ and $f\left(t_{r,g}\right)$ of the mutated individual and the target individual, respectively, and select the one with the smaller objective function to retain. At the same time, consider the party with the smaller ELM weight because it has the better generalization ability.

$$T_{r,g+1} = \begin{cases} T_{trial,r,g}, & if \quad f\left(T_{trial,r,g}\right) \leq f\left(T_{r,g}\right) and \|\beta_{rial,r,g}\| < \|\beta_{r,g}\| \\ T_{r,g}, & otherwise \end{cases} \tag{44}$$

Step 5: Determine if the objective function reaches the optimal value, or if the maximum number of iterations is reached. When the optimal value or the maximum number of iterations is reached, the optimal individual is output. According to the SSADE algorithm above, the optimal input weight $A_i$ and hidden layer bias $b_i$ are searched to obtain a more compact network structure, optimize the ELM algorithm, and improve the performance of the adaptive control item.

### 3.3. Robust Control Algorithm

The main purpose of the robust control term proposed in this paper is to suppress the uncertainty of the model. The residual error is generated by an adaptive control process, and the error $E_{\tau ai}$ of Equation (35) is:

$$E_{\tau ai} = \hat{\tau}_{ai} - \tau_{ai}^* = h_i \hat{\beta}_i - h_i \beta_i^* \tag{45}$$

where $\tau_{ai}^*$ is the optimal approximation of $\tau_{ai}$. Suppose $E_{\tau ai}$ has an upper bound, $\exists E_{\tau ai}^{\max}, \forall |E_{\tau ai}| \leq E_{\tau ai}^{\max}$. The robust controller designed in this paper is as follows.

$$\hat{\tau}_{Ri} = E_{\tau ai}^{max} sign(r_i) \tag{46}$$

where $\hat{\tau}_{Ri}$ represents the estimated output of the robust control term. $r_i$ represents the $i$th term of the systematic error $r$. It can be obtained that the estimated output of the space robot control algorithm is:

$$\hat{\tau}_k = [\hat{\tau}_{k1}, \hat{\tau}_{k2}, \ldots, \hat{\tau}_{ki}, \ldots, \hat{\tau}_{kn}]$$
$$\hat{\tau}_{ki} = M_{ii}(\hat{\tau}_{PDi} + \hat{\tau}_{ai} + \hat{\tau}_{Ri}) = M_{ii}\left(Kr_i + h_i\hat{\beta}_i + E_{\tau ai}^{max}sign(r_i)\right) \tag{47}$$

where $K$ represents the gain of the PD control algorithm.

### 3.4. Stability Analysis

Take the Lyapunov function:

$$V = \frac{1}{2}r^T Mr + \frac{1}{2}\sum_{i=1}^{n}\sum_{j=1}^{l}\frac{1}{\mu_{ij}}\widetilde{\beta}_{ij}\widetilde{\beta}_{ij} \tag{48}$$

Find the first derivative of Equation (48):

$$
\begin{aligned}
\dot{V} &= r^T M\dot{r} + \tfrac{1}{2}r^T \dot{M}r + \sum_{i=1}^{n}\sum_{j=1}^{l}\tfrac{1}{\mu_{ij}}\widetilde{\beta}_{ij}\dot{\hat{\beta}}_{ij} \\
&= r^T\left(M(\ddot{q}_d + \Lambda\dot{e}) + C(\dot{q}_d + \Lambda e) + d - \tau_k - Cr\right) + \tfrac{1}{2}r^T\dot{M}r + \sum_{i=1}^{n}\sum_{j=1}^{l}\tfrac{1}{\mu_{ij}}\widetilde{\beta}_{ij}\dot{\hat{\beta}}_{ij} \\
&= r^T\left(M(\ddot{q}_d + \Lambda\dot{e}) + C(\dot{q}_d + \Lambda e) + d - Kr - \sum_{i=1}^{n}h_i\hat{\beta}_i - E_{\tau ai}^{max}sign(r) - Cr\right) + \tfrac{1}{2}r^T\dot{M}r + \sum_{i=1}^{n}\sum_{j=1}^{l}\tfrac{1}{\mu_{ij}}\widetilde{\beta}_{ij}\dot{\hat{\beta}}_{ij} \\
&= r^T\left(\sum_{i=1}^{n}h_i\beta_i - \sum_{i=1}^{n}h_i\hat{\beta}_i - Kr - E_{\tau ai}^{max}sign(r) - Cr\right) + \tfrac{1}{2}r^T\dot{M}r + \sum_{i=1}^{n}\sum_{j=1}^{l}\tfrac{1}{\mu_{ij}}\widetilde{\beta}_{ij}\dot{\hat{\beta}}_{ij} \\
&= r^T\left(\sum_{i=1}^{n}h_i\beta_i^* - \sum_{i=1}^{n}h_i\hat{\beta}_i - Kr - E_{\tau ai}^{max}sign(r) - Cr\right) + r^T\sum_{i=1}^{n}E_{\tau ai} - \tfrac{1}{2}r^T(2C - \dot{M})r + \sum_{i=1}^{n}\sum_{j=1}^{l}\tfrac{1}{\mu_{ij}}\widetilde{\beta}_{ij}\dot{\hat{\beta}}_{ij} \\
&= -\sum_{i=1}^{n}h_i\widetilde{\beta}_i r_i - r^T Kr - E_{\tau ai}^{max}r^T sign(r) + r^T\sum_{i=1}^{n}E_{\tau ai} + \sum_{i=1}^{n}\sum_{j=1}^{l}\tfrac{1}{\mu_{ij}}\widetilde{\beta}_{ij}\dot{\hat{\beta}}_{ij} \\
&= -r^T Kr - E_{\tau ai}^{max}r^T sign(r) + r^T\sum_{i=1}^{n}E_{\tau ai} \leq -r^T Kr - \sum_{i=1}^{n}\left(E_{\tau ai}^{max} - E_{\tau ai}\right)|r_i| \leq 0
\end{aligned}
\tag{49}
$$

It can be proved that the control algorithm constructed in this paper is stable.

## 4. Simulation

### 4.1. Parameter Settings

The space manipulator structure model designed by the modeling software NX in this paper is shown in Figure 5. From the model, the inertia parameters shown in Table 1 can be obtained by choosing similar materials and structures to the experimental system. Other system parameters shown in Table 1 are set according to the physical parameters of the hardware of the experimental system below.

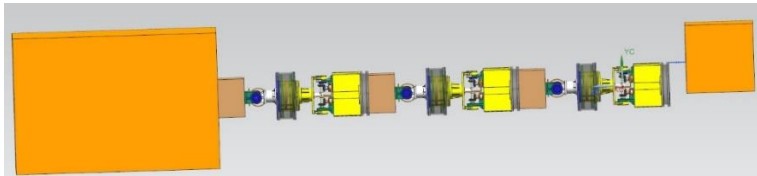

**Figure 5.** The space manipulator structure model.

**Table 1.** The dynamic parameters of the space robot system.

| Description | Symbol | Value | Unit |
|---|---|---|---|
| Base mass | $m_0$ | 100 | kg |
| Joints mass | $m_1, m_2, m_3$ | 5 | kg |
| Base inertia | $J_0$ | 6.67 | kg·m$^2$ |
| Joints inertia | $J_1, J_2, J_3$ | 0.2 | kg·m$^2$ |
| Target mass | $m_t$ | 5 | kg |
| Target inertia | $J_t$ | 2.5 | kg·m$^2$ |
| Vector from center of the base to the centroid of the first joint | $b_0$ | 0.5 | m |
| Vectors from center of the $i$th joint to the centroid of the $i$th body | $a_1, a_2, a_3$ | 0.5 | m |
| Vectors from the centroid of the $i$th body to the center of the $(i + 1)$th joint | $b_1, b_2, b_3$ | 0.5 | m |
| Vectors from center of the Target to the centroid of the end | $a_t$ | 0.1 | m |
| Initial base angle | $\theta_0$ | 0 | rad |
| Initial base angular velocity | $\omega_0$ | 0 | rad/s |
| Initial base linear velocity | $v_0$ | 0 | m/s |
| Initial joint angle | $q_i, i = 1, 2, 3$ | [0, 0, 0] | rad |
| Initial joint angular velocity | $\dot{q}_i, i = 1, 2, 3$ | [0, 0, 0] | rad/s |

In order to verify the validity of the method, this paper uses the model of a planar three-DOFs free-floating space robot as established by the Matlab/Simulink software. The target initial angular velocity is 0.5 rad/s. The interior disturbance term $\boldsymbol{d}$ in the robotic system is set by the function $\boldsymbol{d} = 0.5 + \boldsymbol{rand}(0, 1) \times \boldsymbol{sign}(\dot{\boldsymbol{q}})$. Table 2 shows the control algorithm parameters of the space robot system.

**Table 2.** The parameters of the adaptive control algorithm.

| Algorithm | Description | Symbol | Value |
|---|---|---|---|
| Strategy Self-Adaptation Differential Evolution–Extreme Learning Machine (SSADE–ELM) | Amount of hidden layer nodes of ELM network | $l$ | 60 |
| | Max iteration | $t_{max}$ | 1000 |
| | Population size | $NP$ | 30 |
| | Stage control factor | c | 0.85 |
| | Arbitrary non-zero vector | $\dot{\xi}$ | [1, 1, 1] |
| | Upper bound of error | $E_{\tau ai}^{max}$ | 1 |
| | The gain of the PD control algorithm | K | Diag (100, 50) |
| | Positive diagonal coefficient matrix | $\Lambda$ | Diag (2, 1) |
| Extreme Learning Machine (ELM) | Amount of hidden layer nodes of ELM network | $l$ | 60 |
| | Max iteration | $t_{max}$ | 1000 |
| | Population size | $NP$ | 30 |
| | Input weight | $A_i$ | rand (0, 1) |
| | Hidden layer bias | $b_i$ | rand (0, 1) |
| | Arbitrary non-zero vector | $\dot{\xi}$ | [1, 1, 1] |
| | Upper bound of error | $E_{\tau ai}^{max}$ | 1 |
| | The gain of the PD control algorithm | K | Diag (100, 50) |
| | Positive diagonal coefficient matrix | $\Lambda$ | Diag (2, 1) |
| Particle Swarm Optimization–ELM (PSO–ELM) | Amount of hidden layer nodes of ELM network | $l$ | 60 |
| | Max iteration | $t_{max}$ | 1000 |
| | Population size | $NP$ | 30 |
| | The weights of the stochastic acceleration terms | $c_1 = c_2$ | 0.2 |
| | The inertial weight serving as a tradeoff between the global and local exploration capabilities of the swarm | $w$ | 2 |
| | Arbitrary non-zero vector | $\dot{\xi}$ | [1, 1, 1] |
| | Upper bound of error | $E_{\tau ai}^{max}$ | 1 |
| | The gain of the PD control algorithm | K | Diag (100, 50) |
| | Positive diagonal coefficient matrix | $\Lambda$ | Diag (2, 1) |

*4.2. Simulation Results*

Figure 6 shows the total angular momentum the space robot with target.

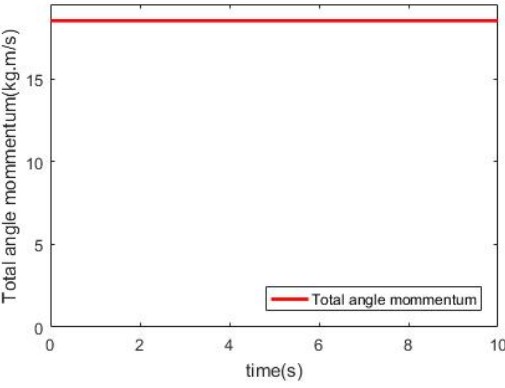

**Figure 6.** Total angular momentum.

Figure 7 shows the space robot attitude parameters by simulation. Figure 7a shows the base angle. Figure 7b shows the base angular velocity. Figure 7c,d show the joints angle via VFF–RLS and RLS. Figure 7e,f show the joints angular velocity via VFF–RLS and RLS.

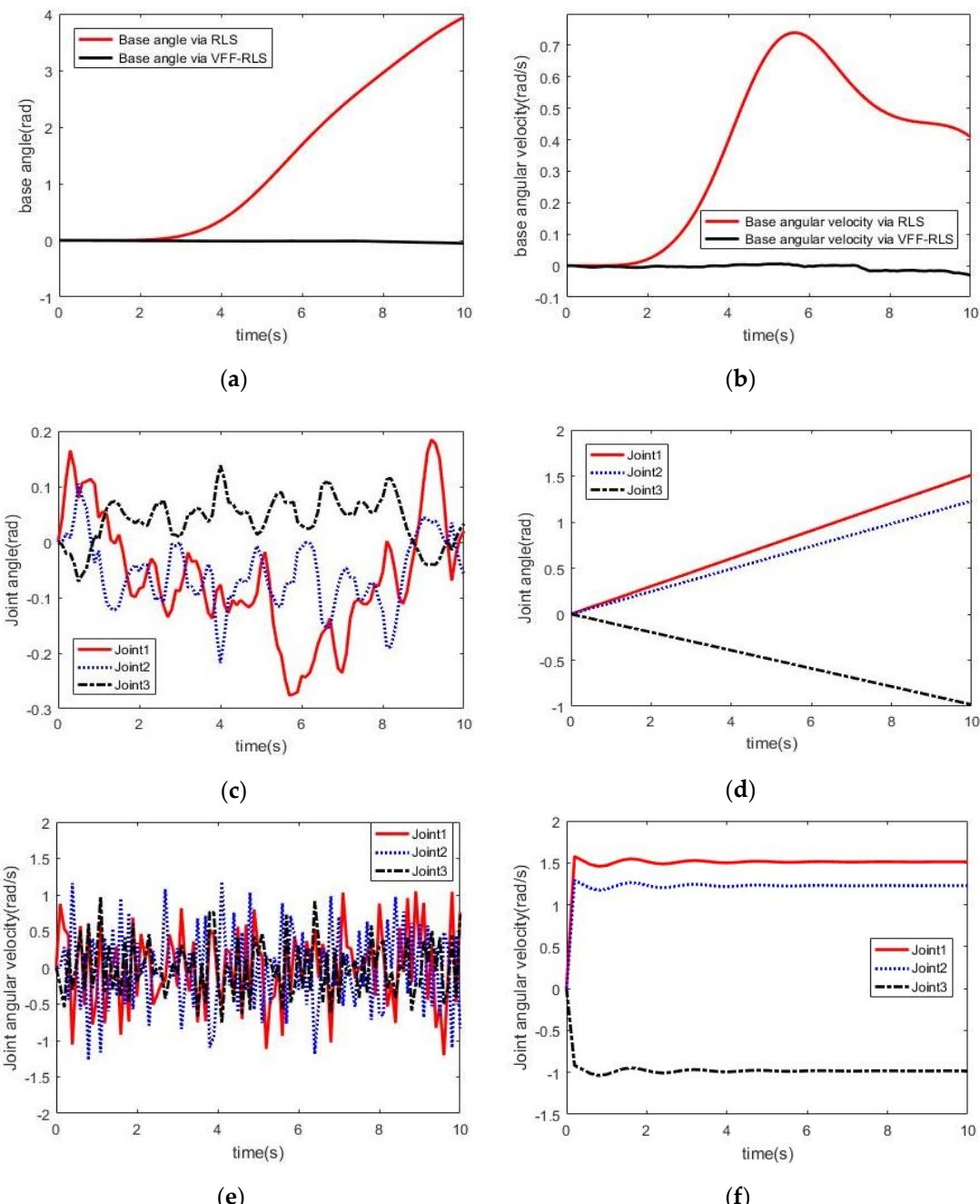

**Figure 7.** Space robot attitude parameters by simulation. (**a**) The base angle; (**b**) The base angular velocity; (**c**) The joints angle via Variable Forgetting Factor Recursive Least Squares (VFF–RLS); (**d**) The joints angle via RLS; (**e**) The joints angular velocity via VFF–RLS; (**f**) The joints angular velocity via RLS.

Figure 8 shows the tracking error of each joint trajectory of control algorithms by simulation. Table 3 shows the average error of joint tracking of control algorithms by simulation.

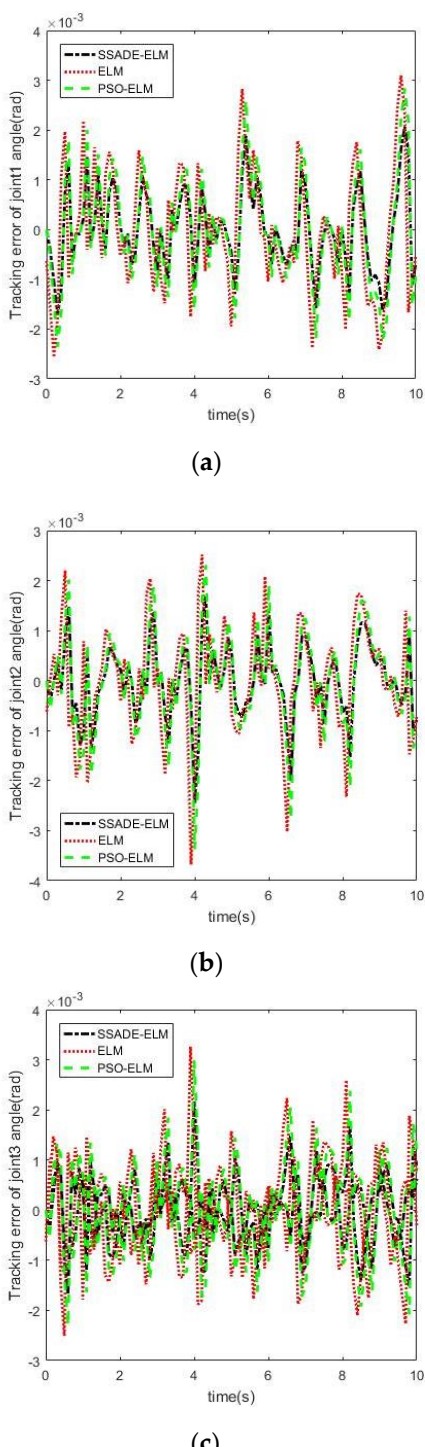

**Figure 8.** The tracking error of each joint trajectory of control algorithms by simulation. (**a**) The tracking error of Joint1 trajectory of control algorithms; (**b**) The tracking error of Joint2 trajectory of control algorithms; (**c**) The tracking error of Joint3 trajectory of control algorithms.

**Table 3.** The average error of joint tracking of control algorithms by simulation. (SSADE–ELM = Strategy Self-Adaptation Differential Evolution–Extreme Learning Machine, PSO–ELM = Particle Swarm Optimization–ELM).

| Algorithm | SSADE–ELM | ELM | PSO–ELM |
|---|---|---|---|
| **Average error** | $-3.0 \times 10^{-7}$ | $-4.5 \times 10^{-7}$ | $-4.2 \times 10^{-7}$ |

## 5. Experiment

### 5.1. Experimental Setting

Based on the parameters in Table 1, a semi-physical experiment system of a space robot principle prototype is constructed. The main hardware architecture includes: a computer for data processing, an FPGA control card as the core control module, and a space robot prototype. The communication module of the joint control system adopts CAN bus. The prototype is placed on an air floating platform supported by air feet and has three yaw freedoms. It is constructed mainly by the joint motors (Kollmorgen TBM(S)–12955-X) and the six-dimensional force sensors (ATI–Nano17) are used for measuring the axial forces and moments. The encoders (EAC58P) are used for measuring the angular displacement and speed. The motor drives (HAR–5/60) are used for controlling motor motion. There is another stiffness motor which we closed for a constant stiffness in this experiment. The diagram of the experimental system is shown in Figure 9. The workflow of the semi-physical experiment system is as follows: the operator operates on the PC and issues commands to the joint control system. The joint control system then sends the command to the motor drive, which rotates to change the joint angle. At the same time, the joint control system uses the motor encoder to monitor the movement of the motor, and the position and speed information of the feedback motor are transmitted to the joint control module and the driver, respectively, and the motor movement is adjusted according to the error, thereby adjusting the movement of the joint to achieve the purpose of movement and precision. Requirements to make the joints have better positioning. The experiment system uses a single pendulum to strike the end of the joint to produce an external disturbance torque $\tau_e$. In order to be able to generate the force in the yaw direction, the experiment was designed to strike the end of the joint at a certain angle to the pitch axis of the joint. The method of calculating the end impact force is as follows:

$$\tau_e = Fsin\theta \times a_t = \frac{m \times (v_0 - v_t)}{\triangle t} sin\theta \times a_t \tag{50}$$

where $F$ is the total collision force, $f_{yaw}$ is the collision force in the yaw direction, m is the mass of the pendulum block, $\triangle t$ is the collision duration, $v_0$ is the initial velocity of the pendulum block during the collision, and $v_t$ is the velocity of the pendulum block after the collision, and we have:

$$v_0 = \sqrt{2gl(1 - \cos\alpha_0)}; v_t = \sqrt{2gl(1 - \cos\alpha_t)} \tag{51}$$

where $g$ is the acceleration of gravity, $l$ is the distance from the fixed end of the inelastic light rope to the centroid of the pendulum block, $\alpha_0$ is the angle between the rope and the vertical direction at the starting position of the pendulum block, and $\alpha_t$ is the angle with the vertical direction when the pendulum block reaches the highest position after the collision. In this experiment, the parameters setting are as follows: $\theta = 30°$, $\alpha_0 = 10°$, $l = 0.2$ m, $g = 9.8$ m $\times$ s$^{-2}$, m $= 5$ kg, $\triangle t = 0.01$ s, the experimental measurements obtain that $\alpha_t \approx 3.8°$, so $F \approx 10$N and $\tau_e \approx 0.5$ N $\cdot$ m.

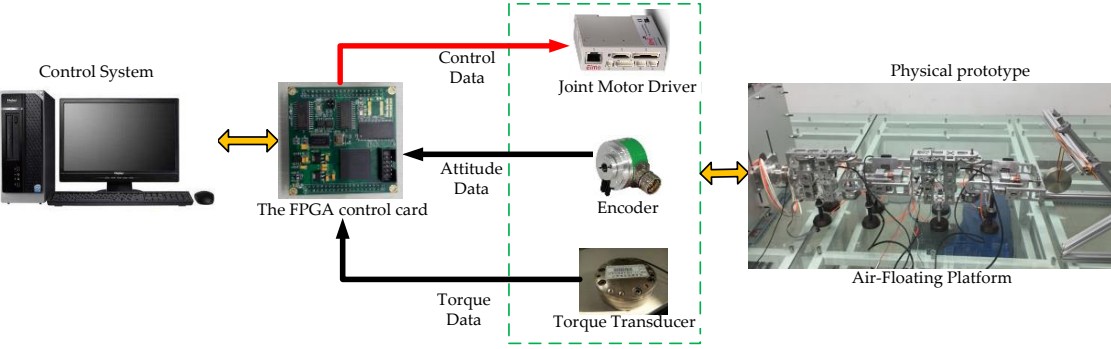

**Figure 9.** The diagram of the experimental system.

The control algorithms used in the experiment system are written by MATLAB. The control algorithm parameter settings are shown in Table 2.

## 5.2. Experimental Results

Figure 10 shows the space robot attitude parameters by experiment. Figure 10a shows the base angle. Figure 10b shows the base angular velocity. Figure 10c,d show the joints angle via VFF–RLS and RLS. Figure 10e,f show the joints angular velocity via VFF–RLS and RLS.

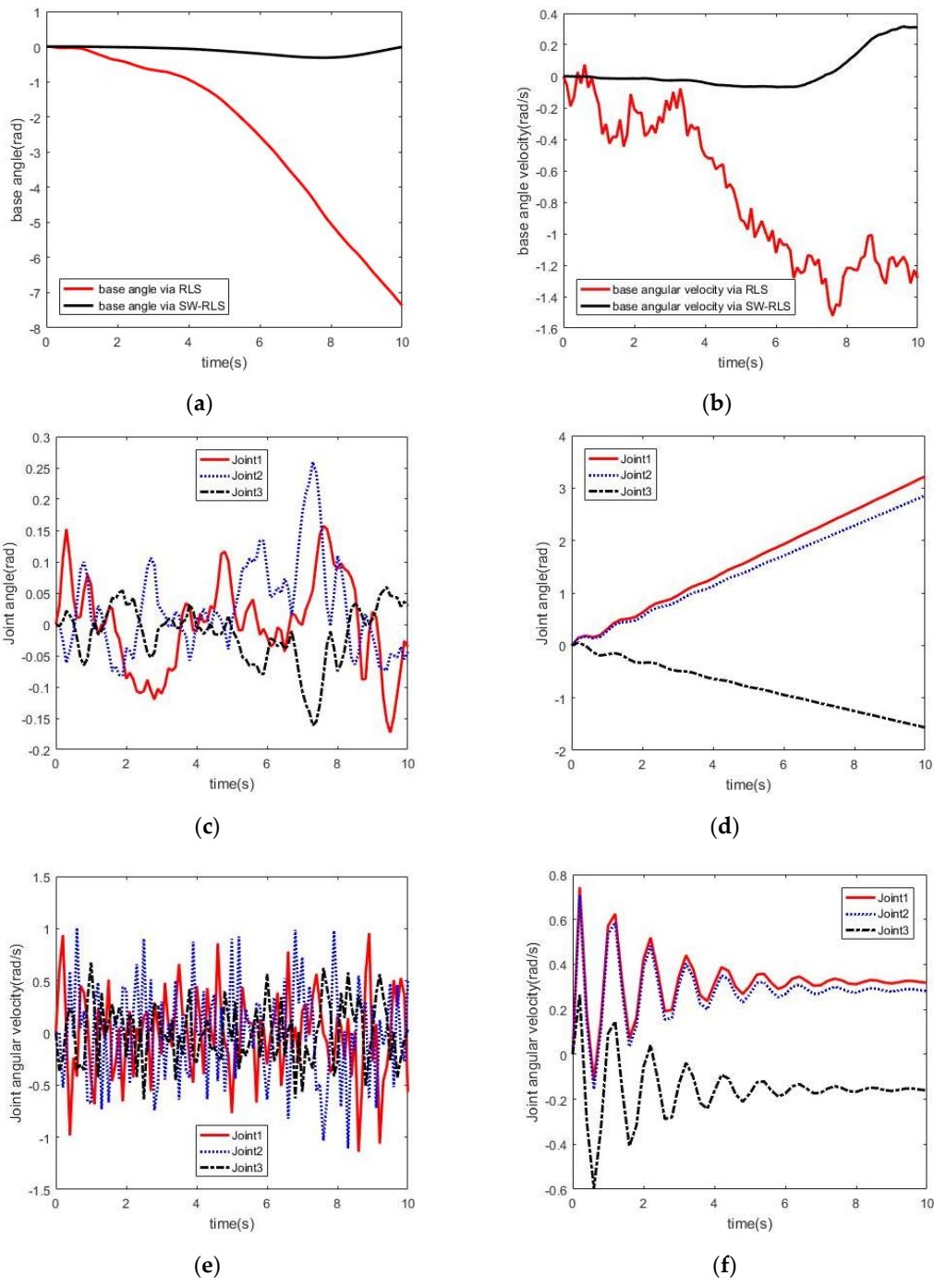

**Figure 10.** Space robot attitude parameters by experiment. (**a**) The base angle; (**b**) The base angular velocity; (**c**) The joints angle via Variable Forgetting Factor Recursive Least Squares (VFF–RLS); (**d**) The joints angle via RLS; (**e**) The joints angular velocity via VFF–RLS; (**f**) The joints angular velocity via RLS.

Figure 11 shows the tracking error of each joint trajectory of control algorithms by experiment. Table 4 shows the average error of joint tracking of control algorithms by experiment.

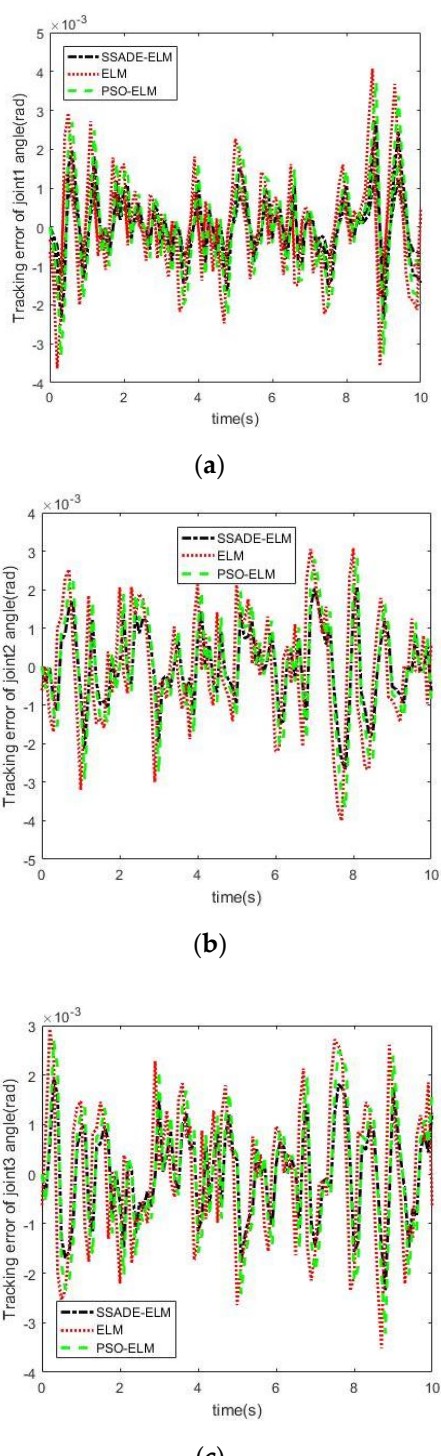

**Figure 11.** The tracking error of each joint trajectory of control algorithms by experiment. (**a**) The tracking error of Joint1 trajectory of control algorithms; (**b**) The tracking error of Joint2 trajectory of control algorithms; (**c**) The tracking error of Joint3 trajectory of control algorithms.

**Table 4.** The average error of joint tracking of control algorithms by experiment. (SSADE–ELM = Strategy Self-Adaptation Differential Evolution–Extreme Learning Machine, PSO–ELM = Particle Swarm Optimization–ELM).

| Algorithm | SSADE–ELM | ELM | PSO–ELM |
|:---:|:---:|:---:|:---:|
| **Average error** | $-5.3 \times 10^{-6}$ | $-7.6 \times 10^{-6}$ | $-6.9 \times 10^{-6}$ |

## 6. Discussion

It can be seen from Figure 6 that when the space robot is in contact with the target, the entire complex system maintains angular momentum conservation. In the simulation, the adaptive reactionless path planning achieves adaptability by using the RLS algorithm and the VFF–RLS algorithm, respectively. It can be seen from Figure 7a,b that the VFF–RLS algorithm can achieve the minimum disturbance planning of the base attitude, but the RLS algorithm cannot. It can be seen from Figure 7d,f that for the system with time-varying characteristics, the RLS algorithm does have a saturation phenomenon, and the time-varying parameters of the tracking system cannot be realized to plan the manipulator attitude parameters. Meanwhile, the VFF–RLS algorithm exhibits good parameter tracking performance for time-varying system, and can realize the time-varying parameters of the tracking system to plan the attitude parameters of the manipulator, thus ensuring the stability of the base attitude. The same results can be obtained in the experiment. It can be seen from Figure 10a,b that the VFF–RLS algorithm can achieve the minimum disturbance planning of the base attitude, but the RLS algorithm cannot, and the offset of the base angle is much bigger than the simulation caused by the unknown disturbance of the semi-physical experiment system. It can be seen from Figure 10d,f that saturation phenomenon of the RLS algorithm makes it unable to plan the manipulator attitude parameters in the time-varying system, while the VFF–RLS algorithm achieves this goal very well. The differences between the manipulator attitude parameters between the simulation and the experiment are also caused by the unknown disturbance of the semi-physical experiment system. The SSADE–ELM algorithm proposed in this paper is compared with the adaptive control algorithm based on ELM and the POS–ELM control algorithm proposed in Reference [21]. As can be seen from Figure 8a–c and Table 3 by simulation, since the adaptive control algorithm based on ELM does not need to perform control parameter optimization, it is superior to the SSADE–ELM control algorithm and the POS–ELM control algorithm based on the trajectory tracking speed. However, in terms of control precision, the SSADE–ELM control algorithm is the best, with the POS–ELM control algorithm following closely. The ELM-based adaptive control algorithm is the worst. This is due to the use of the group intelligence algorithm to optimize the input weight of the ELM algorithm. We can get the same results in the experiment. As can be seen from Figure 11a–c and Table 4, the SSADE–ELM control algorithm has the best trajectory tracking ability, and due to the time overhead of control parameter optimization, the trajectory tracking speeds of the SSADE–ELM control algorithm and the POS–ELM control algorithm are a little slower than the ELM control. Through the above analysis of simulation and experiment, the effectiveness and superiority of the ARNS planning and control algorithm based on SSADE–ELM proposed in this paper are proven.

## 7. Conclusions

An ARNS path planning and control algorithm is proposed for free-floating space robots. The RNS planning strategy is adopted to plan the movement of the robot arm to ensure the stability of the space robot base. At the same time, in order to contact the unknown target, the reactionless motion of the manipulator is realized to ensure the stability of the basic attitude of the space robot, in order to adapt the requirement of time-varying disturbance of the system. The VFF–RLS algorithm is introduced to construct an ARNS path planning algorithm to avoid the saturation phenomenon of the RLS algorithm. This paper proposes a stability control algorithm for space robot contact via the SSADE–ELM algorithm, which can track the dynamic change planning path and improve the speed and accuracy of tracking performance. The proposed algorithm does not require accurate system dynamics modeling, nor does

it require information about unknown time-varying disturbance, enabling dynamic reactionless path tracking control. The simulation results verify the effectiveness and superiority of the proposed algorithm, which makes it important for future on-orbit operation. The algorithm proposed in this paper still has some shortcomings, such as the slow execution speed of the algorithm, which will be the direction of further research.

**Author Contributions:** Project administration, Z.-H.D.; Supervision, J.-C.H.; Writing—original draft, X.Y.

**Funding:** This research was funded by Scientific and technological innovation projects of General Armament Department, grant number ZYX12010001.

**Acknowledgments:** This work was supported by Scientific and technological innovation projects of General Armament Department (grant number ZYX12010001).

**Conflicts of Interest:** The author declares no conflict of interest.

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
