# Peer review of "Research on Adaptive Reaction Null Space Planning and Control Strategy Based on VFF–RLS and SSADE–ELM Algorithm for Free-Floating Space Robot"

_electronics, doi:10.3390/electronics8101111_

Round 1

Reviewer 1 Report

In this paper, the authors investigate the path planning and control problem of free-space robots with unknown targets. An adaptive reaction null space planning strategy is proposed by designing a variable forgetting factor recursive least squares algorithm. And the ELM is employed to address the uncertainties of the system dynamics. Simulation and experimental studies are performed to verify the proposed controller.

   Generally, this paper is interesting. However, the authors should carefully address the following comments.

The equation (49) gives the proof of the semi-negative definite of the Lyapunov function where the neural network (ELM) is employed in the controller design. However, due the approximation error of the neural network, the \dot V could not always converge to (or less than ) zero. Is there should be a small constant (approximation error) in the inequality (49)? Figure 11 shows the tracking performance of the controller. But the significance of the proposed method is not very clear presented. As the green line and the black line is very close. The introduction could be further improved by discussing some more resent literature, for example, about the extreme learning machine, Haptic Identification by ELM Controlled Uncertain Manipulator, IEEE Transactions on Systems, Man, and Cybernetics: Systems; about the null space technique, Mind Control of A Robotic Arm with Visual Fusion Technology, IEEE Transactions on Industrial Informatics; about the robot control, Adaptive Parameter Estimation and Control Design for Robot Manipulators with Finite-Time Convergence, IEEE Transactions on Industrial Electronics. The are many typos in this paper, to list a few,

--in Page 3, the Extreme Learning Machine (ELM)

-- in Page 4, this paper is shown in Figure 1. Where

-- In Page 5, the space robot capture the target

-- In Page 6, space robot complex system obtained by (17) is as

There also some grammatical errors here, for example, “When contact with an unknown target” Please carefully check and revised them.

Reviewer 2 Report

The manuscript proposes a new control strategy based on a reaction null approach for the path planning and control of the arm of a f free-floating space robots with  target.

The formulation reported on the manuscript is formally correct and the numerical and experimental results show that the proposed control algorithms can be used for on-orbit operations

Overall, the manuscript itself is interesting however there are some minor issues that are reported below that require some action from the Authors.

The proposed control algorithms are derived and then tested supposing that no flexible components are present in the manipulator. Neither the links are considered flexible nor the base presents some flexible appendages (like the solar arrays). This hypothesis can be considered reasonable in principle, however it important, when testing the robustness of the proposed controllers, to verify what happens when disturbances coming from the dynamics of the flexible component are present. The mutual coupling among control actions, attitude dynamics and flexibility can un-stabilize the system. I suggest the authors to add some comments about this. In recent years these affects are addressed in some papers (see for instance the papers ‘’ Dynamic/control interactions between flexible orbiting space-robot during grasping, docking and post-docking manoeuvres”, or ‘’ Modeling and observer-based augmented adaptive controlof flexible-joint free-floating space manipulators’’). Other works can be found in recent literature on this topic. Do the Authors verified the performance of the controllers when some uncertainties are considered on the dynamic parameters of the space robot systems (i.e. moment of inertia of the base and of the links or target inertia ?

Author Response

This manuscript is a resubmission of an earlier submission. The following is a list of the peer review reports and author responses from that submission.